# Sample-efficient LLM Optimization with Reset Replay

## Abstract

Recent advancements in LLM post-training, particularly through reinforcement learning and preference optimization, are key to boosting their reasoning capabilities. However, these methods often suffer from low sample efficiency and a susceptibility to primacy bias, a phenomenon where overfitting to initial experiences diminishes network plasticity and damages the learning process. To address these challenges, we introduce LLM optimization with Reset Replay (LoRR), a general and powerful plugin for enhancing sample efficiency in preference-based optimization. Its core mechanism enables high-replay training to maximize the utility of each data batch. To mitigate overfitting, LoRR orchestrates a periodic reset strategy that reuses the initial data and policy to maintain network plasticity, and further adopts a hybrid optimization objective to better exploit training data. Extensive experiments show that LoRR significantly boosts the performance of various preference optimization methods on both mathematical and general reasoning benchmarks. Notably, an iterative DPO framework augmented with LoRR achieves comparable performance on challenging math tasks, rivaling many complex or computationally expensive baselines. Our findings highlight that LoRR offers a practical and sample-efficient paradigm from limited offline data, unlocking greater performance with minimal changes to existing post-training workflows.

## 1. Introduction

Aligning Large Language Models (LLMs) with human intent has become a cornerstone of modern AI, primarily driven by Reinforcement Learning (RL) from human feedback (Leike et al., 2018; Ouyang et al., 2022) and verifiable

[1]Anonymous Institution, Anonymous City, Anonymous Region, Anonymous Country. Correspondence to: Anonymous Author <anon.email@domain.com>.

Preliminary work. Under review by the International Conference on Machine Learning (ICML). Do not distribute.

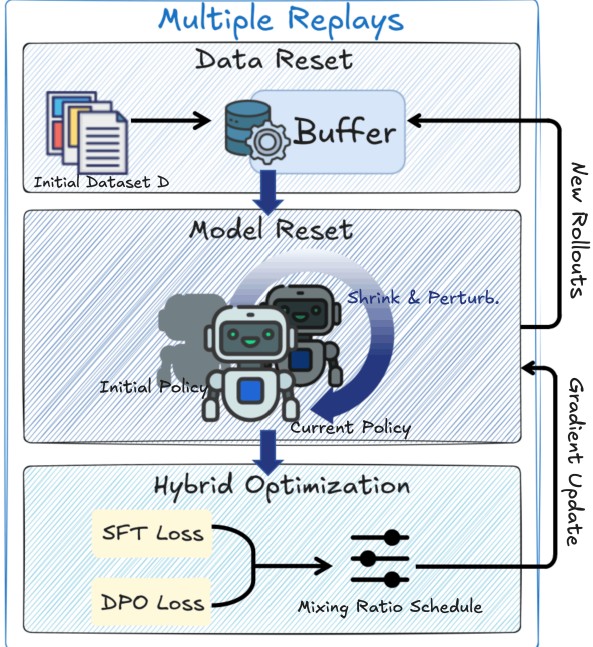

*Figure 1.* The LoRR Framework. To enable high-replay training without collapse, the method coordinates a triple-reset strategy that dynamically balances exploration (via rollouts) and stability (via hybrid optimization) throughout the post-training process.

reward optimization (Cui et al., 2025; Ma et al., 2025). Recently, off-policy methods like Direct Preference Optimization (DPO) (Rafailov et al., 2023) and SimPO (Meng et al., 2024) have streamlined this paradigm by training directly on preference pairs, bypassing the complexity of explicit reward modeling. However, a critical bottleneck in applying off-policy RL algorithms to real-world tasks is their inefficient utilization of offline datasets. While Supervised Fine-Tuning (SFT) can aggressively exploit static data, it often suffers from rapid overfitting and lacks the discriminative power of preference learning (Chu et al., 2025). It creates a fundamental dilemma for RL-based finetuning: relying solely on policy rollouts ensures on-policy consistency but leads to poor sample efficiency; conversely, naively mixing in static offline data to boost memorization introduces severe distribution shifts. Crucially, this shift makes the model susceptible to *primacy bias*—a pathology where the optimizer overfits to initial static experiences (Nikishin et al., 2022), causing a catastrophic loss of plasticity that hinders

learning from subsequent feedback. Consequently, unlocking the potential of offline datasets without triggering this collapse remains an open challenge in LLM post-training.

In the realm of traditional RL, a line of work (Nikishin et al., 2022; D'Oro et al., 2023; Xu et al., 2024) addresses the offline datasets learning efficiency challenge by scaling the number of replays. Empirical evidence suggests that scaling this ratio can remarkably accelerate convergence and boost final performance by squeezing more utility from limited data. However, none of the prior work has made an effort to enable the updating of LLMs at a *high replay number*, meaning no one trains the LLMs multiple times using the collected experiences for periodic interactions with a dataset. When compared with traditional RL, LLM optimization has parameter updates that occur less frequently (Tajwar et al., 2024). As a result, there is a greater chance in LLM that the offline dataset may not be fully utilized, either by SFT-based or RL-based optimization. Given the high computational cost of generating rollout data, this inefficiency creates a compounding bottleneck: if an LLM fails to fully absorb the latent signals from current experiences due to insufficient updates, it struggles to improve its policy rapidly enough to generate higher-quality data in subsequent rounds, thereby stalling the entire self-improvement loop.

To tackle the critical challenge of unlocking the full potential of offline datasets in LLM finetuning, we propose LLM optimization with Reset Replay (LoRR), a versatile plug-and-play module designed to augment any preference optimization. As illustrated in Figure 1, LoRR is designed not as a static combination of techniques, but as a synergistic closed-loop system operating under a high-replay regime. At its core, the framework orchestrates three essential elements to enhance offline data utilization efficiency. First, the *Data Reset* manages the replay buffer by dynamically blending static offline data with fresh policy rollouts, ensuring diverse experiences without incurring severe distribution shifts. To counteract the risk of overfitting inherent in this high-frequency learning, the *Model Reset* periodically applies a shrink & perturb strategy (Ash & Adams, 2020) to inject plasticity, acting as a circuit breaker against primacy bias. Complementing this, *Hybrid Optimization* employs a time-variant schedule to mix SFT and preference losses, serving as a stabilizer to anchor the model to the language manifold during intensive updates. This dynamic equilibrium allows LoRR to sustain effective learning at replay numbers significantly higher ($3\times \sim 5\times$) than standard single-pass limits, turning the risk of overfitting into an opportunity for deeper data exploitation.

Our main contributions are summarized as follows:

- We empirically demonstrate the primacy bias in LLM optimization, and show the dilemma of using diverse experiences in the training process, motivating the need

for mixing rollouts and hybrid optimization.

- We propose LoRR, a plug-and-play module under high-replay training that enhances offline data utilization through a novel synergy of periodic data resetting, plasticity injection, and hybrid loss scheduling.

- Extensive experiments show that LoRR consistently improves performance across various preference optimization methods. Notably, it enables a simple iterative DPO to outperform complex RL-based algorithms in mathematical reasoning tasks while maintaining superior sample efficiency under the same conditions.

## 2. Related Work

**Sample Efficiency for RL**. A central goal in traditional RL is to improve sample efficiency, as acquiring experience through environment interaction is often costly and time-consuming (Chen et al., 2021; D'Oro et al., 2023; Schwarzer et al., 2023). A prominent strategy for achieving this is to maximize the utility of collected data by increasing the replay number, also known as the update-to-data ratio. Despite offering little benefit when applied to standard baselines (Fedus et al., 2020), replay number scaling has been observed to enhance the performance of already well-optimized algorithms. This refers to the number of gradient updates performed on an agent's parameters for each new data point collected (Nikishin et al., 2022; D'Oro et al., 2023). The underlying principle is that an agent can gain more knowledge by repeatedly learning from previously collected experiences, thus reducing the demand for new samples and accelerating learning (Schwarzer et al., 2023). However, while appealing in principle, naively scaling the replay ratio introduces a critical stability issue, overfitting (Nikishin et al., 2022; Lyle et al., 2023; Sokar et al., 2023). From earlier experiences when training, it results in losing the network plasticity to learn good policies in the following learning process. Researchers (D'Oro et al., 2023) further raise the algorithm, using the shrink and perturbation instead of the reset to maintain the network plasticity. While high replay with reset is becoming an active area of investigation in single-/multi-agent RL, this topic remains underexplored within LLM post-training, especially as more and more RL algorithms (Rafailov et al., 2023; Cui et al., 2025) are being applied to LLMs. This gap motivates our work to develop a framework that can harness the benefits of a high replay number while explicitly mitigating the associated risks in the LLM optimization context.

**RL for LLM Reasoning**. RL has been extensively employed in the context of LLMs primarily for aligning models with human preferences (Ouyang et al., 2022; Meng et al., 2024), also in some applications like jailbreaking (Liu et al., 2024). Recently, a growing body of work investi-

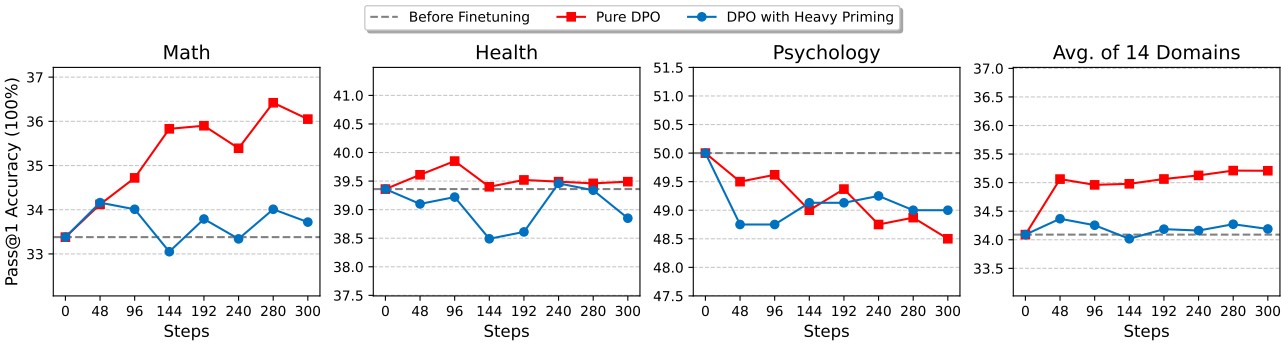

*Figure 2.* DPO training process gains in MMLU-pro (Avg.) and its subdomains. It compares an LLM trained with standard procedures against one subjected to heavy priming on its initial data.

gates the amplification of mathematical reasoning in open-source language models, particularly the Qwen2.5 family, through RL (Zeng et al., 2025; Yan et al., 2025; Ma et al., 2025; Cui et al., 2025). Early efforts (Zeng et al., 2025; Ma et al., 2025) employed explicit, verifiable rewards demonstrating notable performance gains on math benchmarks. Subsequent research (Wang et al., 2025b) focused on data efficiency, enabling effective learning even from minimal labeled or unlabeled examples. However, the broader generalizability of these observed gains, particularly within mathematical domains, has faced critical scrutiny. For example, performance enhancements noted on Qwen2.5, even when driven by random or inaccurate rewards, did not consistently translate to other LLMs, suggesting model-specific idiosyncrasies (Zhao et al., 2025). Consequently, while RL may be served as a potent tool for specialized LLM enhancement, its successful deployment necessitates a nuanced understanding of its interaction with specific learning dynamics. Our work aims to enhance the sample efficiency of RL to improve the general learning ability of LLMs, extending beyond mere mathematical benchmarks.

**Self-improvement of LLMs** primarily occurs during the training phase, where models enhance their performance by evaluating and refining their own outputs without relying on external human supervision (Chu et al., 2025). However, a prevailing view suggests that LLMs cannot achieve self-improvement without any external information (Huang et al., 2025). Consequently, an increasing number of methods, especially RL-based, leverage supervisory signals to facilitate self-improvement, such as outcome labels (Tu et al., 2025; Zhang et al., 2025), value functions (Wang et al., 2024a), and process reward models (Cui et al., 2025). Our approach is based on self-improvement, and it keeps training and partially resets the data and model for each batch.

## 3. The Primacy Bias in LLM Optimization

This section examines how initial training phases can disproportionately affect the LLM finetuning process due to primacy bias. To start, let us review how previous work (Nik-

ishin et al., 2022; Zhou et al., 2022) defined **the primacy bias** in deep reinforcement learning as follows:

**Definition 3.1.** *A tendency to overfit initial experiences that damages the rest of the learning process.*

In the context of LLM post-training, current optimization paradigms often rely on RL or preference-based rewards and collect data via off-policy methods. Typically, a policy periodically rolls out several responses and selects preferences based on a reward verifier. This raises the question of whether primacy bias exists in the finetuning process. To this end, we present two experiments focused on LLM post-training, aiming to illustrate the existence of this phenomenon induced by the improper learning of early-stage data. Firstly, we demonstrate that over-training an LLM exclusively on its initial data can irreparably damage the rest of the learning process. Secondly, our experiments reveal that the applicability of the initial experiences of LLMs is uncertain during the training process.

**Heavy Priming Causes LLM Overfitting**. The extent to which LLM finetuning relies on its initial data affects its learning process. In order to explicitly reveal the impact of primacy bias in LLM finetuning, we investigate an extreme case of over-reliance on early data: *Does overfitting to just one batch of initial data destroy the generalization of an LLM?* To this end, we design a set of experiments to optimize preferences over *math datasets* based on Llama-3B. As shown in Figure 2, we start by using the default hyperparameters for pure DPO (red line), which means each batch gets one update per step. Then, we set up an identical version of DPO with heavy priming mode (blue line). In this setup, we first train the LLM with a small batch 200 times, and then feed the remaining data to the LLM for standard training recovery. The results show that even after training on all data, the LLM affected by heavy priming was unable to recover from the initial overfitting in the math domain. Besides, both normal training and overfitting have more or less impact on other domain-specific performance, such as health and psychology, suggesting that knowledge of the original model is being forgotten (Ibrahim et al., 2024). This finding

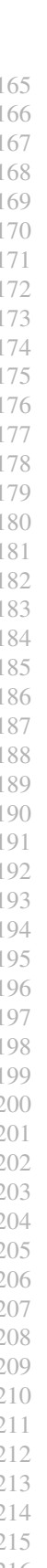

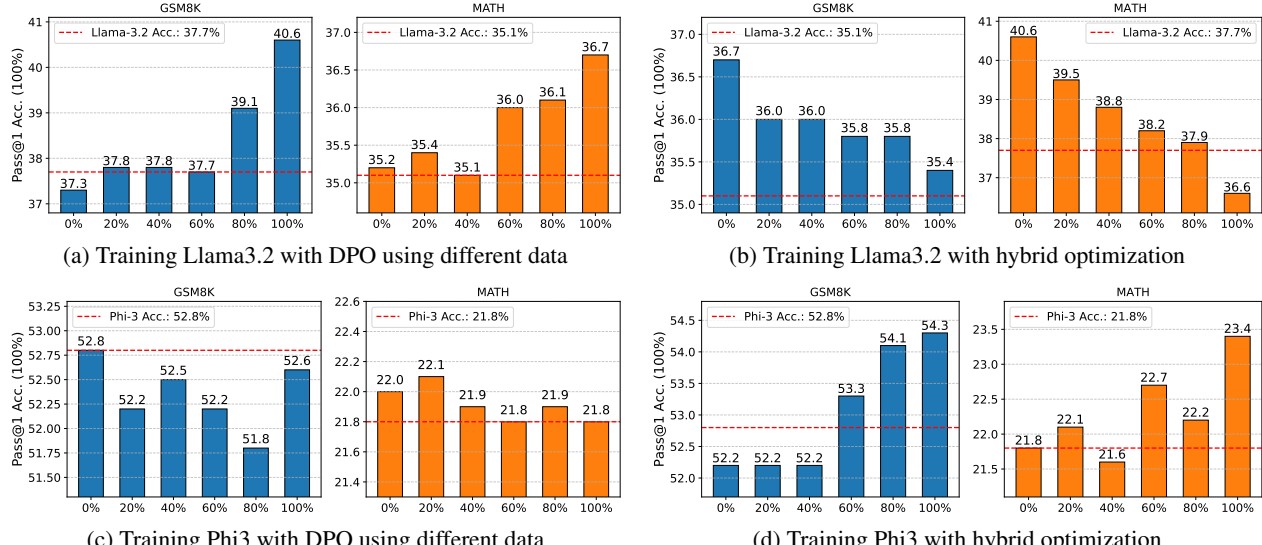

(a) Training Llama3.2 with DPO using different data

(b) Training Llama3.2 with hybrid optimization

(c) Training Phi3 with DPO using different data

(d) Training Phi3 with hybrid optimization

*Figure 3.* DPO training with different ratios of the rollout data (Figs. (a), (c)) and SFT loss (Figs. (b), (d)), where the higher the percentage the more mixed it is. We pick two benchmarks, GSM8K and Math. The red dotted lines are the performance of the base models.

strongly suggests that the primacy bias has compounding effects similar to RL (Nikishin et al., 2022): an LLM that is overfitted early on will rollout poorer quality data, which in turn leads to less efficient learning, further impairing its ability to learn iteratively without reset.

**The Dilemma of Static Data Utilization**. Existing preference learning algorithms typically rely solely on rollout data while ignoring the original responses generated by the base model, which significantly constrains the model's capabilities to the performance bounds of the base model (Ma et al., 2025). However, directly incorporating original data into the optimization process can bring non-trivial challenges: an LLM initially cannot determine whether the collected original data is useful for learning, leading to potential inefficiencies or even degradation in performance. To explore this issue, we try in practice to optimize LLMs by blending different ratios of data, i.e., combining a portion of the original dataset responses with rolling out preference scored by a reward verifier. In addition, we integrate hybrid losses, such as DPO for preference exploration and SFT for answer fitting. As shown in Figure 3, the results reveal a significant inconsistency in data utility. Surprisingly, these experiments reveal model-specific patterns rather than consistent trends. Llama3.2, which appears to have acquired strong math domain capabilities during pre-training, leverages rollout data to generate high-quality samples for further improvement; in contrast, Phi3 requires prior data for SFT fitting, and incorporating initial experience even has a negative impact. These findings highlight that there is no one-size-fits-all static ratio. Instead of relying on fixed mixtures, which lead to unpredictable outcomes, we need a dynamic mechanism to balance the reliance on stable initial experiences against the need for exploiting new, high-reward rollouts.

## 4. Scaling Replay Number with Resets

Inspired by the success of high replay number in traditional RL (D'Oro et al., 2023; Fedus et al., 2020), we propose to adapt this paradigm for LLM finetuning. However, distinct from RL agents, LLMs are more susceptible to primacy bias when updated frequently on limited batches without intervention. To address this gap, we introduce LLM optimization with Reset Replay (LoRR) into the finetuning pipeline. As the name implies, LoRR incorporates triple reset operations (on model, data, and loss) during multi-round replays, a design tailored to mitigate overfitting while maximizing offline data value. *The full details of Algorithm 1 provided in the Appendix A.* Notably, LoRR follows the general workflow of standard on-policy preference optimization but employs two additional operations. The main operation is in line 17 of the pseudocode, resetting the policy model $\pi_\theta$ in replays. Given a reset interval $|\mathcal{B}|$ with $L$ times of replay for each batch $\mathcal{B}$, LoRR performs a reset to inject plasticity into the policy model, thereby restoring the learning capability of specified networks. The second is hybrid optimization in line 20 of pseudocode, which has been shown to work differently for diverse on-policy data (Tajwar et al., 2024; Ma et al., 2025). This augmentation operation with different ratios can take full advantage of LoRR updates, bringing benefits of diversity to input representations for LLM learning. Below, we detail the core operations designed to maintain plasticity and maximize data efficiency.

**Replays with Model Reset**. The model reset technique we integrated is the Shrink & Perturb strategy (Ash & Adams, 2020), which periodically re-initializes a portion of the network parameters to maintain the network plasticity. Originally proposed as a method to "warm-start" neural network

training for incorporating new data without losing generalization, this technique has recently been adopted in RL systems (D'Oro et al., 2023; Lyle et al., 2023; Xu et al., 2024) to combat overfitting in high replay number settings. The formulation for LLM shrink & perturb is defined as

$$\pi_\theta \leftarrow \alpha\pi_\theta + (1 - \alpha)\pi_{\theta_{\text{init}}}, \tag{1}$$

where $\pi_\theta$ is a current LLM policy learned in the finetuning phase and $\pi_{\theta_{\text{init}}}$ is the initial policy. The reset ratio $\alpha$ determines the extent to which the existing LLM parameters are preserved during an update. A higher $\alpha$ indicates a greater influence from the new updates, while a lower $\alpha$ means more of the old parameters are retained.

**Replays with Data Reset**. As preference algorithms (Meng et al., 2024; Tran et al., 2023) collect data using the on-policy setting, we develop a periodic data reset into replays. Specifically, we first sample multiple trajectories $\{\mathbf{y}_1, \cdots, \mathbf{y}_K\}$ for each prompt $\mathbf{x}$, and then re-annotate them with a reward verifier $r$ that selects the highest-scoring one as $\mathbf{y}_w$ and the lowest-scoring one as $\mathbf{y}_l$ (Pace et al., 2024). The preference pairs $(\mathbf{x}, \mathbf{y}_w, \mathbf{y}_l)$ generated by the reward signals are collected in a set of pairs $\mathcal{T}$, a.k.a., a transition buffer. However, to avoid duplicating the update with each replay, a certain ratio $\varepsilon$ of initial experiences is mixed into the transition buffer, where the pair is formalized as $(\mathbf{x}, \mathbf{y}, \mathbf{y}_l')$. $\mathbf{y}_l'$ is from the dataset $\mathcal{D}$ if it has preference data; otherwise, it is rolled out in the initial model $\pi_{\theta_{\text{init}}}$. For every replay $\ell$, we set a linear ratio $\varepsilon$ working on each buffer. The ratio goes up as the number of replays declines, which means the model further learns from the initial experiences. In practice, given an initial ratio $\varepsilon_{\text{init}} = 1$, each replay adjusts the ratio to be $\varepsilon \leftarrow \varepsilon_{\text{init}}(1 - \frac{\ell}{2L})$. This decaying schedule acts as a soft curriculum. In early replays, a higher $\varepsilon$ grounds the model in known high-quality demonstrations to prevent early collapse. As $\ell$ increases, the model gradually shifts focus to its own generated rollouts, allowing it to refine its policy on the newly explored high-reward regions.

**Replays with Hybrid Optimization**. Both SFT and preference learning methods operate within the same optimal policy-reward subspace, positioning SFT as a special case of implicit reward learning (Wang et al., 2025a). Beyond the preference optimization, RL with SFT has been shown to increase capacity in LLM finetuning (Ma et al., 2025). In §3 we have explored the uncertainty of LLM experience, but what about preference optimization combined with SFT? Thus, in high replays, we directly mix different proportions of SFT losses, allowing the policy model to learn $\mathbf{y}_w$ directly. Mathematically, given a preference loss[1] $\mathcal{L}_{\text{DPO}}$ and an SFT loss $\mathcal{L}_{\text{SFT}}$, the hybrid way in each replay is

$$\mathcal{L}_\theta = \lambda\mathcal{L}_{\text{SFT}} + (1 - \lambda)\mathcal{L}_{\text{DPO}}, \tag{2}$$

---

[1]Our default optimizing loss is DPO, but this can be replaced at will, depending on scenarios.

where $\theta$ denotes the parameters of current policy $\pi_\theta$ and $\lambda$ is an SFT ratio. The rollout response $\mathbf{y}_w$ naturally aligns with higher reward signals throughout iterations; hence, as the number of replays increases, so does our $\lambda$. Formally, we set an initial value $\lambda_{\text{init}} = 0$ that the first optimization is based on preferences only, and then for each optimization after a reset, the SFT ratio adjusts as $\lambda \leftarrow \frac{\ell}{2L} + \lambda_{\text{init}}$. As this ratio grows, it was theoretically proven (Tajwar et al., 2024) that the policy distribution slowly aligns with the expected distribution. And thanks to reset strategies, the hybrid loss slows down the primary bias in LLM optimization.

**Why LoRR Works?** Rather than a mere combination of techniques, LoRR operates as a synergistic closed-loop system: *High Replay* is essential for maximizing data extraction but risks rapid plasticity loss; the *Triple Reset* counteracts this rigidity by periodically restoring the optimization landscape; crucially, *Hybrid Optimization* acts as a stabilizer against the potential manifold collapse caused by resets. This dynamic equilibrium allows LoRR to scale update frequencies significantly beyond standard limits without overfitting. Next, we validate this effectiveness across extensive math and reasoning benchmarks.

## 5. Experiments

In this section, we mainly evaluate LoRR on mathematical and reasoning problems, highlighting the superior performance of LoRR as a plugin with different preference methods (§5.2). We then investigate the performance of LoRR on iterative DPO and compare it to existing RL-/SFT-based methods with multi-round improvements, analyzing potential differences in reasoning paradigms (§5.3). Further, we provide comprehensive ablation experiments in §5.4.

### 5.1. Experimental Setup

**Datasets**. Our training set is augmented with mathematical problems from a subset of Meta-Math (Yu et al., 2024) and MMIQC (Liu et al., 2025). This data has been processed into high-quality preference data by previous work (Lai et al., 2024) and used for finetuning various mathematical models. To ensure that the amount of data in §5.3 is consistent with the baseline (e.g., DPO-VP (Tu et al., 2025)), the same 8K data are sampled for training. In addition, we optimize on a universal dataset, UltraFeedback (Cui et al., 2023), for comparing the reasoning performance of models.

**Models and Finetuning Implementation**. We first optimize preferences with three model families: `Llama3.2-3B` (Grattafiori et al., 2024), `Phi3-mini-4K` (Abdin et al., 2024), and `Qwen2.5-3B` (Yang et al., 2025), all in the Instruct setup. For a fair comparison, we use the same reward model as SimPO (Meng et al., 2024): `ArmoRM-Llama3-8B-v0.1` (Wang et al.,

*Table 1.* Llama3.2 finetuning results on six mathematical benchmarks. We train SFT models on the 8K math data for base, iterative, and LoRR settings. For Instruct settings, we use off-the-shelf models as the SFT model.

| Training Method | GSM8k | MATH | TabMWP | Minerva | AMC23 | AIME24 | **Avg.** |
|---|---|---|---|---|---|---|---|
| Llama3.2-3B-Instruct | 37.7 | 35.1 | 27.5 | 33.8 | 12.5 | 6.7 | 25.55 |
| Llama3.2 + DPO (Base) | 40.6 | 36.7 | 27.3 | 33.4 | 15.0 | 3.3 | 26.05 |
| + DPO (Iter. n) | 41.9 | 36.1 | 28.1 | 35.2 | 15.0 | 6.7 | 27.2 |
| + DPO (Base+LoRR) | 43.0 | 37.9 | 28.2 | 36.0 | 15.0 | **10.0** | 28.35 |
| Llama3.2 + KTO (Base) | 39.0 | 36.1 | 27.3 | 35.2 | 17.5 | 6.7 | 26.97 |
| + KTO (Iter. n) | 39.4 | 36.2 | 26.8 | 32.4 | 20.0 | 10.0 | 27.47 |
| + KTO (Base+LoRR) | 40.5 | 37.3 | 28.8 | 36.2 | 20.0 | 10.0 | 28.80 |
| Llama3.2 + IPO (Base) | 37.8 | 36.0 | 26.7 | 33.6 | 15.0 | 6.7 | 25.97 |
| + IPO (Iter. n) | 40.9 | 36.9 | 27.7 | 35.4 | 12.5 | 6.7 | 26.70 |
| + IPO (Base+LoRR) | 40.3 | 37.1 | 28.9 | 36.0 | 12.5 | 6.7 | 26.92 |
| Llama3.2 + rDPO (Base) | 37.7 | 35.9 | 27.6 | 33.4 | 15.0 | 3.3 | 25.48 |
| + rDPO (Iter. n) | 40.6 | 37.8 | 29.5 | 36.6 | 12.5 | 6.7 | 27.28 |
| + rDPO (Base+LoRR) | 40.9 | 38.1 | 28.4 | **39.4** | 20.0 | 6.7 | 28.92 |
| Llama3.2 + SimPO (Base) | 39.0 | 36.8 | 28.4 | 33.8 | 12.5 | 3.3 | 25.63 |
| + SimPO (Iter. n) | 40.9 | 37.6 | 26.4 | 34.4 | 15.0 | 6.7 | 26.83 |
| + SimPO (Base+LoRR) | **45.7** | **38.6** | **32.3** | **39.4** | 20.0 | 3.3 | **29.88** |

2024b) for ranking generated data, significantly enhancing performance. We test the performance of LoRR under five different sets of preference optimizations: DPO (Rafailov et al., 2023), KTO (Ethayarajh et al., 2024), IPO (Azar et al., 2024), rDPO (Park et al., 2024), and SimPO (Meng et al., 2024), where the hyperparameters remain unchanged and are marked as *Base* in tables. For LoRR, we set the number of replays to 3 and integrate in these preference optimizations to see the performance of LoRR. To ensure a rigorous comparison, we add an iterative training style baseline (*Iter. n*) as a standard online preference learning loop, where the number of iterations is 3, and each iteration rolls data out through the reward model. Crucially, distinct from LoRR, this baseline does not perform parameter or data resets; it represents the standard continuous learning paradigm where model weights are cumulatively updated, making it susceptible to the primacy bias discussed in §3.

**Evaluation** In our evaluation, we mainly focus on several well-established mathematical benchmarks: GSM8K (Cobbe et al., 2021), MATH/MATH500 (Hendrycks et al., 2021), TabMWP (Lu et al., 2023), Minerva (Lewkowycz et al., 2022), OlympiadBench (He et al., 2024), and AIME/AMC (Li et al., 2024). All of the benchmarks adopt Pass@1 as the evaluation criterion. Considering reasoning experiments training on UltraFeedback, we further evaluate the generalization ability on MMLU-Pro (Wang et al., 2024c), where we randomize the order of multiple-choice options to avoid contamination.

### 5.2. Main Results

Our initial evaluation focuses on quantifying the performance improvement of the LoRR plugin on different preference optimizations. The experimental procedure involves finetuning models on the previously mentioned 8K math dataset, first using standard preference learning to create baseline models, and then enhancing the process with LoRR. Results of these base preference learning and LoRR-based versions for the math ability are shown in Table 1, and Tables 5 and 6 in Appendix C. As we can see, the introduction of LoRR leads to a performance boost, far exceeding the results of the base optimization methods. Specifically, the different finetuning models achieve max 6.54% to 16.99% enhancements relative to the SFT model, using the LoRR setting. It also produces a significant boosting effect compared to methods where the interaction with the dataset is fixed (e.g., the number of iterations is 3). Some optimizations (e.g., training Llama3.2 with rDPO) were originally trained to cause side effects, yet even these cases outperform baselines when augmented with LoRR. In a side-by-side comparison, SimPO has the most significant performance effect when plugged in LoRR, maintaining first-place performance on all three models, followed by rDPO. Note that the small sample size of AIME24 resulted in no significant enhancement, possibly because the SFT data is sampled by $\pi_{\theta_{init}}$, but this policy has no resolution capability in its knowledge. Overall, LoRR is able to enhance most of the scenarios under different optimizations.

To evaluate the broader applicability of LoRR, we extend our experiments to a general domain. We optimize the Llama3.2 on the UltraFeedback dataset and subsequently test on the challenging MMLU-Pro reasoning task. As baselines, we employ two different preference optimization algorithms, DPO and SimPO, and each algorithm incorporates a different contribution of SFT loss. We then augmented these baseline methods with LoRR to isolate its impact. Figure 4 illustrates the comparative performance, highlighting the improvements conferred by our method. Specifically, it shows that varying the proportion of SFT loss yields minimal performance improvements and induces overfitting after approximately 400 training steps. In stark contrast, LoRR

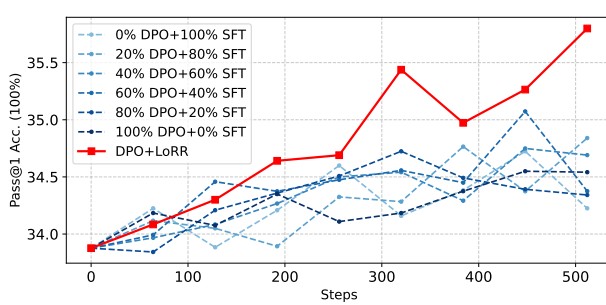 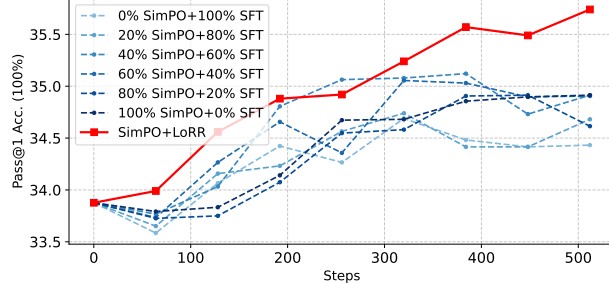

(a) Training with DPO on UltraFeedback      (b) Training with SimPO on UltraFeedback

*Figure 4.* Average pass@1 accuracy of fine-tuned LLama3.2 models on MMLU-Pro's 14 domain test benchmarks. Since different replays of LoRR change the SFT ratio, we select preference learning to mix varying ratios of SFT losses as baselines.

achieves a significant performance increase by efficiently reusing sample data through its reset-and-replay cycle.

### 5.3. Results on iterative DPO

Further, we implement LoRR using an iterative DPO (a.k.a., multi-round self-improvement DPO) framework applied to the Qwen2.5-Math-7B backbone (Yang et al., 2024), which is consistent with most works (Guan et al., 2025; Cui et al., 2025; Zeng et al., 2025). Our training protocol spans three iterations. At the conclusion of each iteration, we perform dynamic rollouts with the current policy $\pi_\theta$ to synthesize new preference pairs. This newly generated preference data is labeled by a reward model, making it closer to an on-policy setting, as discussed in (Meng et al., 2024). Note that we do not use rule-based reward signals because LoRR can train on general datasets, not just for answer matching. For the resulting policy in each iteration, we name it as *Qwen2.5-7B-DPO-LoRR-itern*, demonstrating the progressive gains of this iterative refinement.

To create a rigorous benchmark against RL-based and SFT-based methods, we evaluate our method alongside eleven state-of-the-art baselines (detailed in Appendix D) on standard benchmarks including GSM8K (Cobbe et al., 2021), MATH (Hendrycks et al., 2021), and complex Olympiad-level tasks such as AMC23, AIME24, and Olympiad-Bench (He et al., 2024). A major challenge in fair evaluation is the variance in training data and reward model configurations across prior works. To address this and align with the "zero" setting, the responses **y** are generated by the initialization policy rather than retrieved from static datasets. Specifically, we utilize the same set of 8K math questions mentioned previously to train LoRR, ensuring a comparable data regime to the primary RL baselines. The comparison with pure SFT methods using less expert data is using the same with the authors, e.g., LIMO (Ye et al., 2025) and S1 (Muennighoff et al., 2025). Table 2 reports the Pass@1 accuracy across these methods. For clarity, we categorize the results into two groups: The first section lists reference models where direct comparison is confounded by differ-

ences in training data scale; the second section presents a relatively fair comparison among methods operating under similar constraints, e.g., 8k data source and training epochs. Additionally, we provide Table 7 to analyze the computational cost, detailing the training data consumption and GPU usage differences among the competing methods.

Based on the results in Table 2 and Table 7 (in Appendix D), we draw the following conclusions:

- **Trained with LoRR, a simple iterative DPO achieves mathematical reasoning that is comparable to RL-based and SFT-based methods.** Our final model, Qwen2.5-7B-DPO-LoRR, attains an average score of 47.8, the highest among all methods in the relatively fair comparison group. This performance not only exceeds other DPO variants like Qwen2.5-7B-DPO-VP (46.9) but also outperforms prominent RL-based models such as Simple-RL-Zero (45.7), all while using the same base model and initial training data constraints.

- **The iterative application of DPO demonstrates consistent and significant performance improvements.** We observe a clear monotonic progression in the average score across iterations: from 42.1 in the first iteration, to 46.2 in the second, and culminating in 47.8 in the third. This highlights the effectiveness of our reset replays for data reusing.

- **This superior performance is achieved with remarkable data and computational efficiency, presenting a highly practical and scalable paradigm.** Our method requires only 8K preference pairs for the entire iterative training process, and the entire training for our model was completed in just 2 days on a single A100. This is a fraction of the data used by other top-performing models like Qwen2.5-7B-ReLIFT (45K) and orders of magnitude less than rStar-Math-7B ($\sim$3.6M) with more GPUs, demonstrating our approach's ability to learn effectively from limited data.

In summary, our work demonstrates that LoRR plugin an iterative DPO can achieve leading performance in mathemat-

*Table 2.* Pass@1 results across different methods on math tasks. All the models have been fine-tuned based on the Qwen2.5-Math-7B. The results with † are the ones we evaluate from the published models or retrain via official repositories, and the rStar results with ∗ are obtained from (Guan et al., 2025). In the relatively fair comparison, **Bold** indicates the best results and underline indicates the second one.

| Benchmark | Base | Non-comparable Baselines | | | | | Comparable Baselines (zero setting or less expert data) | | | | | | DPO-LoRR | | |
|---|---|---|---|---|---|---|---|---|---|---|---|---|---|---|---|
| | Qwen2.5† | Instruct† | rStar* | PRIME† | ReLIFT† | LUFFY† | LIMO† | S1† | Simple-RL† | PURE-VR† | DPO-R1† | DPO-VP† | iter1 | iter2 | iter3 |
| MATH500 | 66.2 | 84.6 | 78.4 | 74.0 | 81.4 | 80.6 | 67.0 | 72.6 | **77.8** | 72.8 | 75.4 | 76.0 | 73.8 | 75.4 | 75.6 |
| Minerva | 10.7 | 37.1 | - | 39.7 | 23.9 | 30.5 | 22.8 | 26.1 | 32.7 | 13.6 | 27.6 | 30.9 | 23.5 | 33.8 | **36.0** |
| Olybench | 24.1 | 39.9 | 47.1 | 35.6 | 44.3 | 45.6 | 32.3 | 35.1 | **40.7** | 28.1 | 38.8 | 38.2 | 35.1 | 37.0 | 37.2 |
| AMC23 | 47.5 | 62.5 | 47.5 | 57.5 | 62.5 | 72.5 | 42.5 | 52.5 | 57.5 | 50.0 | 60.0 | **62.5** | 55.0 | 55.0 | 60.0 |
| AIME24 | 23.3 | 16.7 | 26.7 | 23.3 | 13.3 | 13.3 | 16.7 | 20.0 | 20.0 | 23.3 | 23.3 | 26.7 | 23.3 | **30.0** | **30.0** |
| **Avg.** | 34.4 | 48.2 | - | 46.0 | 45.1 | 48.5 | 36.3 | 41.3 | 45.7 | 37.6 | 45.0 | 46.9 | 42.1 | 46.2 | **47.8** |

*Table 3.* Sensitivity analyses on hyperparameters and projection modules. We report the Pass@1 accuracy on GSM8k and MATH. For ratios $\alpha$ and max $\lambda$, we sweep values from 0.0 to 1.0; and for the projection, we evaluate different module configurations independently.

| Parameter | 0.0 | | 0.25 | | 0.5 | | 0.75 | | 1.0 | |
|---|---|---|---|---|---|---|---|---|---|---|
| | GSM8k | MATH | GSM8k | MATH | GSM8k | MATH | GSM8k | MATH | GSM8k | MATH |
| Alpha ($\alpha$) | 38.3 | 35.3 | 42.1 | 36.5 | 43.0 | 37.9 | 43.6 | 36.1 | 41.7 | 36.1 |
| Lambda ($\lambda$) | 42.1 | 37.3 | 42.2 | 36.8 | 43.0 | 38.0 | 43.2 | 37.6 | 42.4 | 36.9 |

| **Projection** | full_layers | | down_proj | | up_proj | | o_proj | | lm_head | |
|---|---|---|---|---|---|---|---|---|---|---|
| | GSM8k | MATH | GSM8k | MATH | GSM8k | MATH | GSM8k | MATH | GSM8k | MATH |
| Projection | 37.8 | 36.5 | 42.8 | 38.3 | 39.3 | 36.3 | 43.0 | 37.9 | 40.5 | 36.4 |

ical reasoning without the need for massive-scale SFT data, complex reward modeling pipelines, or extensive computational clusters, offering a powerful and accessible alternative to prevailing RL-based finetuning methodologies.

## 5.4. Ablation

To fully validate the effectiveness of each core component in the proposed LoRR and clarify the impact of key hyperparameters on performance, we conduct comprehensive sensitivity analyses and ablation experiments. We also investigate whether LoRR on different Qwen3 series model sizes can keep plasticity in Appendix E. All experiments are consistent with the main experiment.

The sensitivity analyses are illustrated in Table 3. First, we observe that the extreme values of $\alpha$ yield suboptimal results. Setting $\alpha \to 0$ (aggressive reset towards initialization) disrupts the learning trajectory, preventing the effective accumulation of knowledge, as evidenced by the significant drop to 38.3% on GSM8k. Conversely, the optimal performance is achieved in the range of $[0.5, 0.75]$, confirming that a moderate injection of plasticity best preserves the model's generative capabilities while allowing for continued optimization. Next, regarding the SFT ratio $\lambda$ in the hybrid loss, the results favor a balanced approach over unimodal objectives, although gains are not significant. While pure preference learning ($\lambda = 0.0$) and pure SFT ($\lambda = 1.0$) achieve decent baselines, the hybrid configuration ($\lambda \approx 0.5$) consistently outperforms both. Then, we investigate the impact of applying the Shrink & Perturb reset to specific architectural components. Interestingly, applying the reset globally to full_layers proves detrimental, likely due

to the destruction of learned features essential for reasoning. Instead, targeted resets on specific components yield superior returns. The o_proj modules emerge as the most effective target for plasticity injection, which is also a default setting. Further, we also provide an in-depth understanding of each component and replay number in the Appendix E.

## 6. Conclusion

In this paper, we introduced the LLM optimization with Reset Replay (LoRR), to address the critical issues of low sample efficiency and primacy bias in LLM post-training. Our algorithm is a general-purpose plugin that combines high-replay training with periodic parameter resets and a hybrid optimization loss, enabling more effective data utilization. Extensive experiments show that LoRR significantly enhances various preference optimization methods on both mathematical and general reasoning tasks. Notably, an iterative DPO approach augmented with LoRR achieves state-of-the-art performance, outperforming complex RL-based methods with substantially greater sample and computational efficiency. This work demonstrates that by mitigating overfitting from intensive data reuse, LoRR unlocks the latent potential in training data, offering a practical path toward more powerful and efficient LLM optimization. As for future work, there are two key aspects. On one hand, a higher replay number (e.g., $L = 10$) tends to become less effective in the later stages of training; therefore, it may be feasible to address this issue through adaptive replay frequencies and reset strength. On the other hand, it is crucial to further understand and reveal the potential mechanisms of plasticity in LLM networks when they are fine-tuned.

## Impact Statements

This paper presents work whose goal is to advance the field of machine learning. There are many potential societal consequences of our work, none of which we feel must be specifically highlighted here.

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

## A. Pseudo Code

---

**Algorithm 1** LLM Optimization with Reset Replay

---

1: **Input:** Language model $\pi_{\theta_{\text{init}}}$, initial preference tuple $(\mathbf{x}, \mathbf{y}, \mathbf{y}')$ from the dataset $\mathcal{D}$, reward verifier $r$, sample number $K$, replay number $L$, reset ratio $\alpha$, SFT ratio $\lambda_{\text{init}}$, rollout ratio $\varepsilon_{\text{init}}$, and total iteration $N$

2: Initialize policy model $\pi_\theta \leftarrow \pi_{\theta_{\text{init}}}$, reference model $\pi_{\text{ref}} \leftarrow \pi_{\theta_{\text{init}}}$

3: **for** each iteration $n \leftarrow 1$ to $N$ **do**

4:     Sample batch of prompts $\mathcal{B} \sim \mathcal{D}$

5:     **for** each replay $\ell \leftarrow 1$ to $L$ **do**

6:         Adjust rollout ratio $\varepsilon \leftarrow \text{Adj}(\varepsilon_{\text{init}}, \ell)$ , and SFT ratio $\lambda \leftarrow \text{Adj}(\lambda_{\text{init}}, \ell)$, initialize $\mathcal{T} \sim \{\}$

7:         **for** each prompt $\mathbf{x} \in \mathcal{B}$ **do**

8:             Generate $K$ responses $\{\mathbf{y}_1, \cdots, \mathbf{y}_K\} \sim \pi_\theta(\cdot|\mathbf{x})$ {rollout new preference data}

9:             Compute outcome rewards $r_i = r(\mathbf{x}, \mathbf{y}_i)$ for $i \in \{1, \cdots, K\}$

10:            Choose the best $\mathbf{y}_w = \arg\max_{\mathbf{y}_i \in \{\mathbf{y}_1, \cdots, \mathbf{y}_K\}} r_i$, and the worst $\mathbf{y}_l = \arg\min_{\mathbf{y}_i \in \{\mathbf{y}_1, \cdots, \mathbf{y}_K\}} r_i$,

11:            **if** $\varepsilon < \epsilon$, $\epsilon \sim U(0, 1)$ **then**

12:                Add reannotated samples to $\mathcal{T} \leftarrow \mathcal{T} \cup \{(\mathbf{x}, \mathbf{y}_w, \mathbf{y}_l)\}$ {update experiences for the transition buffer}

13:            **else**

14:                Add initial samples to $\mathcal{T} \leftarrow \mathcal{T} \cup \{(\mathbf{x}, \mathbf{y}, \mathbf{y}'_l)\}$ {reset data by initial experiences}

15:            **end if**

16:         **end for**

17:         Reset policy model $\pi_\theta \leftarrow \alpha\pi_\theta + (1 - \alpha)\pi_{\theta_{\text{init}}}$ {reset model by shrink and perturbation}

18:         Update policy model $\pi_\theta$ by hybrid losses on $\mathcal{T}$:

19:         $\mathcal{L}_{\text{SFT}} = -\mathbb{E}_{(\mathbf{x}, \mathbf{y}_w, \mathbf{y}_l) \sim \mathcal{T}}\left[\log \pi_\theta(\mathbf{y}_w|\mathbf{x})\right]$, $\mathcal{L}_{\text{DPO}} = -\mathbb{E}_{(\mathbf{x}, \mathbf{y}_w, \mathbf{y}_l) \sim \mathcal{T}}\left[\log \sigma \left(\beta \log \frac{\pi_\theta(\mathbf{y}_w|\mathbf{x})}{\pi_{\text{ref}}(\mathbf{y}_w|\mathbf{x})} - \beta \log \frac{\pi_\theta(\mathbf{y}_l|\mathbf{x})}{\pi_{\text{ref}}(\mathbf{y}_l|\mathbf{x})}\right)\right]$

20:         $\mathcal{L}_\theta = \lambda\mathcal{L}_{\text{SFT}} + (1 - \lambda)\mathcal{L}_{\text{DPO}}$ {hybrid optimization during replays}

21:         Update reference model $\pi_{\text{ref}} \leftarrow \pi_\theta$

22:     **end for**

23: **end for**

24: **Output:** Optimized policy model $\pi_\theta$

---

## B. Implementation Details

In this section, we outline the specific parameters and data of the experiments.

**General training hyperparameters**. We conduct preliminary experiments for the basic preference learning settings using a batch size of 128 and a single training epoch. Moreover, we set the maximum sequence length at 2048 and utilize the AdamW optimizer with a cosine learning schedule, which includes 10% warmup steps, for the preference optimization. For each preference method, we follow the most parameterised exploration of SimPO (Meng et al., 2024), where Table 4 shows the detailed hyperparameters for baselines. For LoRR, we set the default parameters: a replay number $L = 3$, a sample number $K = 5$, and an iteration number $N = 7$ for math. For LoRR in iterative DPO, $N = 3$ is set, and 3 epochs are guaranteed, while other parameters remain unchanged. For all the tasks, we set the reset ratio $\alpha$ to 0.5 and apply it to output projections of networks for the Shrink & Perturb strategy. Eight NVIDIA A100 GPUs are employed to train models, with DeepSpeed ZeRO2 (equipped with CPU offloading) utilized to mitigate GPU memory overhead during training.

**Building datasets for optimization**. For math tasks, our training dataset is supplemented with mathematical problems selected from subsets of Meta-Math (Yu et al., 2024) and MMIQC (Liu et al., 2025). To maintain consistency in data volume with the baselines as discussed in §5.3, we sampled the same 8K data points for training. For the reasoning tasks, we conduct optimizations on a general-purpose dataset, UltraFeedback (Cui et al., 2023), to facilitate comparisons of the models' reasoning capabilities. Regarding training data, we produce a maximum of 5 responses for each prompt in each replay. Additionally, we leverage the ArmoRM model (Wang et al., 2024b) to label the preference relationships between these responses.

*Table 4.* Various preference optimization hyperparameters used for each training setting.

| Method | $\beta$ | $\gamma$ | Learning rate |
|--------|------|------|---------------|
| DPO | 0.01 | - | 5.0e-7 |
| KTO | 0.01 | 1.0 | 5.0e-7 |
| IPO | - | 0.5 | 5.0e-7 |
| rDPO | 0.01 | 0.6 | 5.0e-7 |
| SimPO | 2 | 0.55 | 1e-6 |

*Table 5.* Phi3 finetuning results on six mathematical benchmarks. We train SFT models on the 8K math data for base, iterative, and LoRR settings, where the training modes include DPO, KTO, IPO, rDPO, and SimPO. For Instruct settings, we use off-the-shelf models as the SFT model.

| Training Method | GSM8k | MATH | TabMWP | Minerva | AMC23 | AIME24 | **Avg.** |
|-----------------|-------|------|--------|---------|-------|--------|----------|
| Phi3-mini-4K-Instruct | 52.8 | 21.8 | 19.2 | 23.4 | 20.0 | 3.3 | 23.42 |
| Phi3 + DPO (Base) | 52.6 | 21.8 | 18.7 | 24.4 | 12.5 | 3.3 | 22.22 |
| + DPO (Iter. n) | 51.9 | 19.9 | 17.2 | 18.2 | 22.5 | 3.3 | 22.17 |
| + DPO (Base+LoRR) | 53.9 | 23.8 | 17.6 | **25.2** | 15.0 | **6.7** | 23.70 |
| Phi3 + KTO (Base) | 52.8 | 21.6 | 19.0 | 23.8 | 17.5 | 3.3 | 23.00 |
| + KTO (Iter. n) | 51.3 | 21.1 | 18.9 | 18.2 | 20.0 | 3.3 | 22.13 |
| + KTO (Base+LoRR) | **54.2** | 23.8 | 19.7 | 24.2 | 20.0 | 3.3 | 24.20 |
| Phi3 + IPO (Base) | 49.7 | 20.2 | 16.6 | 18.4 | 15.0 | 3.3 | 20.53 |
| + IPO (Iter. n) | 51.9 | 21.1 | 19.5 | 20.2 | 20.0 | 3.3 | 22.67 |
| + IPO (Base+LoRR) | 51.9 | 21.7 | 19.7 | 19.8 | 17.5 | **6.7** | 22.88 |
| Phi3 + rDPO (Base) | 52.8 | 21.7 | **19.8** | 20.6 | 20.0 | 0.0 | 22.48 |
| + rDPO (Iter. n) | 50.2 | 20.6 | 17.7 | 19.8 | 17.5 | 3.3 | 21.52 |
| + rDPO (Base+LoRR) | 52.5 | 22.4 | 18.1 | 23.6 | **25.0** | 3.3 | 24.15 |
| Phi3 + SimPO (Base) | 51.7 | 21.2 | 19.5 | 22.2 | 17.5 | 3.3 | 22.57 |
| + SimPO (Iter. n) | 48.7 | 19.2 | 16.1 | 18.6 | 17.5 | **6.7** | 21.13 |
| + SimPO (Base+LoRR) | 53.2 | **24.2** | 19.0 | 24.0 | 22.5 | **6.7** | **24.93** |

# C. Additional Experiment Results

We further evaluate the performance of the LoRR plugin on Phi3 and Qwen2.5 models, with results presented in Tables 5 and 6 respectively. These experiments follow the same setup as the Llama3.2 evaluation: finetuning on the 8K math dataset using standard preference learning (baseline) and LoRR-enhanced preference learning, with performance measured across the six mathematical benchmarks. Similarly to the body text, the LoRR-based versions consistently outperform their base preference learning and iterative counterparts. The results for Phi3 and Qwen2.5 reinforce the effectiveness of LoRR across different model architectures, consistently enhancing performance across most mathematical benchmarks when integrated with various preference optimization techniques.

*Table 6.* Qwen2.5 finetuning results on six mathematical benchmarks. We train SFT models on the 8K math data for base, iterative, and LoRR settings, where the training modes include DPO, KTO, IPO, rDPO, and SimPO. For Instruct settings, we use off-the-shelf models as the SFT model.

| Training Method | GSM8k | MATH | TabMWP | Minerva | AMC23 | AIME24 | **Avg.** |
|---|---|---|---|---|---|---|---|
| Qwen2.5-3B-Instruct | 41.1 | 60.4 | 32.1 | 60.8 | 25.0 | 3.3 | 37.12 |
| Qwen2.5 + DPO (Base) | 44.3 | 60.9 | 32.4 | 60.2 | 25.0 | 0.0 | 37.13 |
| + DPO (Iter. n) | 42.5 | 60.8 | 30.5 | 60.0 | 27.5 | 0.0 | 36.88 |
| + DPO (Base+LoRR) | **45.4** | **61.1** | 32.5 | 61.2 | 27.5 | 0.0 | 37.95 |
| Qwen2.5 + KTO (Base) | 44.3 | 60.5 | 29.6 | 60.6 | 22.5 | 0.0 | 36.25 |
| + KTO (Iter. n) | 43.1 | 60.6 | 32.3 | 60.2 | 27.5 | 0.0 | 37.28 |
| + KTO (Base+LoRR) | 44.3 | 59.8 | 30.9 | 60.4 | 27.5 | 3.3 | 37.70 |
| Qwen2.5 + IPO (Base) | 43.4 | 60.3 | 31.2 | 61.4 | 22.5 | 3.3 | 37.02 |
| + IPO (Iter. n) | 42.3 | 60.0 | 28.1 | 60.2 | 30.0 | 0.0 | 36.77 |
| + IPO (Base+LoRR) | 44.0 | 60.3 | 31.1 | **61.8** | 27.5 | **6.7** | 38.57 |
| Qwen2.5 + rDPO (Base) | 44.5 | 60.8 | 29.6 | 61.4 | 22.5 | 3.3 | 37.02 |
| + rDPO (Iter. n) | 43.1 | 60.6 | 32.3 | 61.2 | 27.5 | 0.0 | 37.45 |
| + rDPO (Base+LoRR) | 44.9 | **61.1** | 30.3 | 58.0 | 32.5 | 3.3 | 38.35 |
| Qwen2.5 + SimPO (Base) | 41.8 | 60.6 | 29.6 | 61.0 | 27.5 | 3.3 | 37.30 |
| + SimPO (Iter. n) | 37.7 | 57.4 | 28.4 | 58.6 | 32.5 | **6.7** | 36.88 |
| + SimPO (Base+LoRR) | 41.8 | 60.4 | **32.9** | **61.8** | **35.0** | **6.7** | **39.77** |

# D. Supplements to Iterative DPO

Here, we introduce some baselines based on RL and iterative DPO methods for our comparison, and compare their efficiency in learning offline data in Table 7.

- Qwen2.5-Math-7B-Instruct (Yang et al., 2024): The instruction-tuned base model from the Qwen family, selected for its strong mathematical reasoning capabilities.

- rStar-Math-7B (Guan et al., 2025): A model trained on a large volume of data generated via self-evolution with Monte Carlo Tree Search, which is then filtered using a Process Preference Model.

- Eurus-2-7B-PRIME (Cui et al., 2025): A model trained using policy rollouts and outcome labels, leveraging implicit process rewards.

- Qwen2.5-7B-LIMO (Ye et al., 2025) and Qwen2.5-7B-S1 (Muennighoff et al., 2025) are pure SFT methods using less expert data.

- Simple-RL-Zero (Zeng et al., 2025) and Qwen2.5-7B-PURE-VR (Cheng et al., 2025): Two similar models, trained by RL with verifiable rewards and some tricks, were also trained without additional SFT data.

- Qwen2.5-7B-ReLIFT (Ma et al., 2025) and Qwen2.5-7B-LUFFY (Yan et al., 2025): Two models are both reinforcement learning frameworks that bridge the gap between RL and SFT by incorporating off-policy reasoning traces into the training process.

- Qwen2.5-DPO-R1-Zero (Zhang et al., 2025) and Qwen2.5-7B-DPO-VP (Tu et al., 2025): Both investigate the effectiveness of iterative DPO in facilitating self-improvement for LLMs.

*Table 7.* A comparison of data and GPUs among different methods.

| | Qwen2.5-Math-7B-Instruct | rStar-Math-7B | Eurus-2-7B-PRIME | Qwen2.5-7B-SimpleRL-Zero | Qwen2.5-7B-ReLIFT | Qwen2.5-7B-DPO-VP | Qwen2.5-7B-DPO-LoRR |
|---|---|---|---|---|---|---|---|
| Paradigm | RL + ORM | MCTS + PPM | RL + PRM | RL + VR | RL + SFT | DPO + VR | DPO + SFT |
| Base Model | Qwen2.5-Math-7B | Qwen2.5-Math-7B | Qwen2.5-Math-7B | Qwen2.5-Math-7B | Qwen2.5-Math-7B | Qwen2.5-Math-7B | Qwen2.5-Math-7B |
| SFT Data | 2.5M (open-source and in-house) | ~7.3M (MATH, NuminaMath, etc.) | 230K | 0 | 0 | 0 | 0 |
| RM Data | 618K (in-house) | ~7K (in-house) | 0 | 0 | 0 | 0 | 0 |
| RM | Qwen2.5-Math-RM (72B) | 7B PPM | Eurus-2-7B-SFT | Rule-based RM | Rule-based RM | DeepSeek-V3 | ArmoRM-Llama3-8B-v0.1 |
| Improvement Data | 66K | ~3.647M | 150K | 8K | 45k | 8K | 8K |
| Epoch | - | 2 | - | 1 | 3 | 6 | 3 |
| GPUs / Time | - | 80 H100 / few days | 8 A100 / few days | 32 H100 / 1.5 days | 16 A800 / - | 1 A100 / 3 days | 1 A100 / 2 days |

# E. Ablation Study and Analysis

To comprehensively validate the effectiveness of LoRR, we further conduct a series of ablation studies focusing on its core components, the impact of replay frequency, and its scalability across different model sizes.

**Contribution of Core Components.** We first disentangle the contributions of data reset and hybrid optimization. We compare the full LoRR algorithm against two ablation settings applied to DPO: (i) *Data Reset Only*, where we adjust the rollout ratio $\varepsilon$ but exclude the hybrid loss (i.e., $\lambda = 0$); and (ii) *Hybrid Optimization Only*, where we adjust the SFT ratio $\lambda$ in training but sample training data exclusively from current policy rollouts. As summarized in Figure 5, the results demonstrate that both components contribute distinct performance gains over the pure DPO baseline. Specifically, while the *Data Reset* strategy provides stability, its performance relies heavily on the quality of the initial policy's rollouts. In contrast, *Hybrid Optimization* significantly improves sample efficiency during reset replays by directly leveraging high-reward samples ($\mathbf{y}_w$), resulting in more robust improvements. The full LoRR combines these strengths to achieve optimal performance.

**Sensitivity to Replay Number.** We also explore the effect of a high number of replays, visualizing the performance of LoRR under different $L$ values on the six math benchmarks, as presented in Figure 6. Consistent with observations in traditional RL (D'Oro et al., 2023; Xu et al., 2024), we find that setting $L \in \{2, 3, 5\}$ drastically improves sample efficiency compared to standard single-pass updates ($L = 1$). This finding mirrors observations in traditional RL agents (D'Oro et al., 2023; Xu et al., 2024), but marks the first validation of this phenomenon in LLM finetuning. However, pushing the replay number to an extreme (e.g., $L = 10$) leads to performance degradation even with resets. We attribute this decline to overfitting on the specific batch experience: excessive reuse of collected data causes the policy to collapse onto the current samples, thereby losing the plasticity required to generate diverse, high-quality rollouts in subsequent steps. Regarding the optimal setting, while $L = 5$ yields marginally higher scores than $L = 3$, it incurs a 66% increase in training time. We also hypothesize that for tasks with higher ambiguity (non-math domains), the optimal $L$ might shift lower to avoid fitting reward noise. Balancing computational efficiency, stability, and performance, we adopt $L = 3$ as the default for LoRR.

**Scalability across Model Sizes.** A critical question is whether larger architectures, which typically possess stronger representation capabilities, inherently mitigate the need for the explicit plasticity injection provided by LoRR. To address this, we extended our evaluation to the Qwen3 family (Yang et al., 2025), spanning a wide range of parameters from 0.6B to 14B. The results are summarized in Table 8. Contrary to the hypothesis that larger models possess sufficient intrinsic resistance to overfitting, our findings indicate that LoRR delivers consistent and often substantial gains across the spectrum. Most notably, on the largest model evaluated (e.g., Qwen3-14B), LoRR achieves a remarkable improvement, boosting the average accuracy from 48.0% (standard DPO) to 60.5%. (LoRR+DPO) This empirical evidence suggests that "primacy bias" and the loss of plasticity are not merely artifacts of limited capacity in smaller models (e.g., 0.6B, where LoRR also yields a +7.6% gain), but are fundamental challenges in high-replay iterative finetuning. Even for larger models, the *Triple Resets* effectively prevent the policy from collapsing into suboptimal local minima, enabling continuous improvement.

**Conclusion.** These analyses affirm that standard preference learning often fails to effectively utilize gathered data due to plasticity loss. LoRR, conversely, activates the latent potential of on-policy data through an efficient, scalable replay strategy.

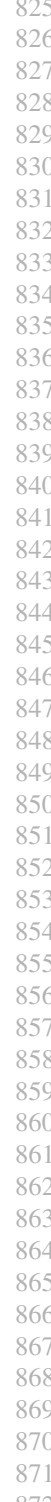
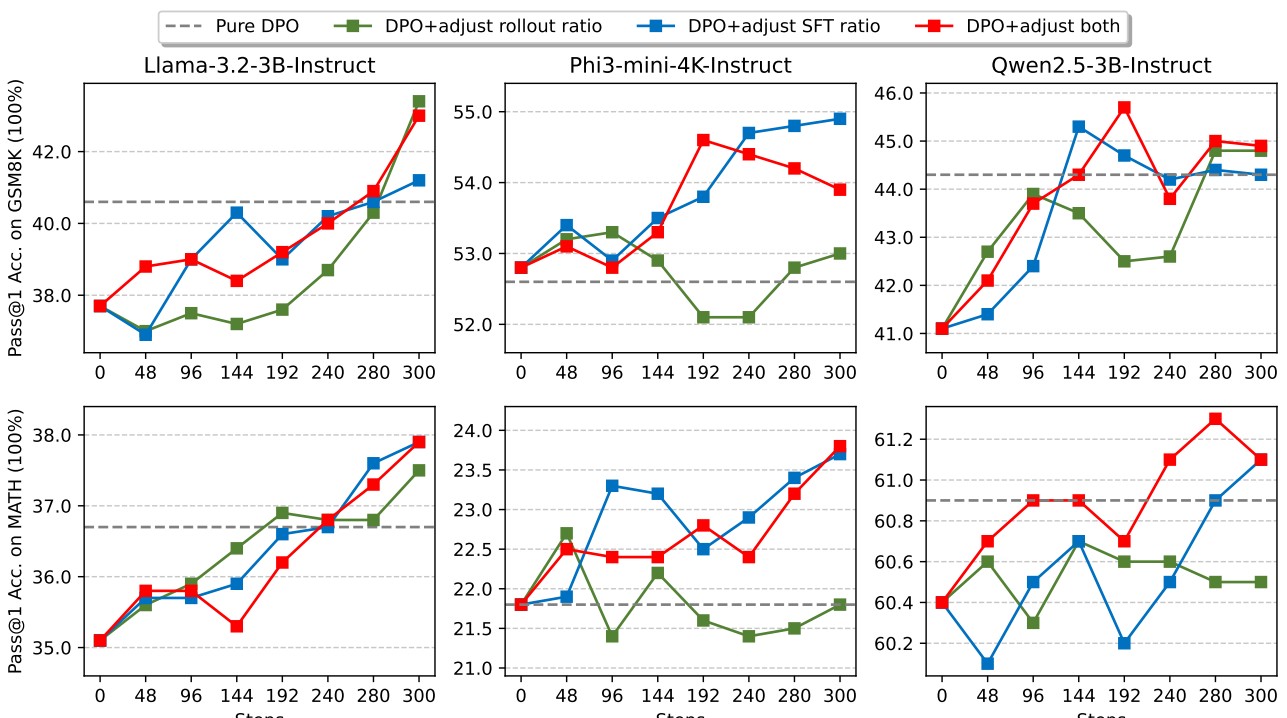

*Figure 5.* Comparison of different components of LoRR. The experiments were fine-tuned on each of the three language models with DPO and tested on GSM8K and MATH tasks.

*Table 8.* Performance comparison of Qwen3 series (Yang et al., 2025) on GSM8k and Math benchmarks.

| Model Size | Qwen3-0.6B | | | Qwen3-1.7B | | | Qwen3-4B | | | Qwen3-8B | | | Qwen3-14B | | |
|---|---|---|---|---|---|---|---|---|---|---|---|---|---|---|---|
| | Base | DPO | LoRR+DPO | Base | DPO | LoRR+DPO | Base | DPO | LoRR+DPO | Base | DPO | LoRR+DPO | Base | DPO | LoRR+DPO |
| GSM8k | 7.3 | 17.7 | 24.3 | 19.0 | 51.4 | 49.1 | 52.2 | 75.6 | 68.9 | 22.5 | 39.7 | 42.8 | 64.3 | 51.5 | 69.3 |
| MATH | 20.5 | 27.5 | 36.1 | 35.5 | 38.4 | 39.5 | 22.6 | 32.8 | 43.2 | 25.4 | 47.2 | 39.4 | 15.9 | 44.5 | 54.7 |
| Avg. | 13.9 | 22.6 | 30.2 | 27.2 | 44.9 | 44.3 | 37.4 | 54.2 | 56.1 | 23.9 | 43.5 | 41.1 | 40.1 | 48.0 | 60.5 |

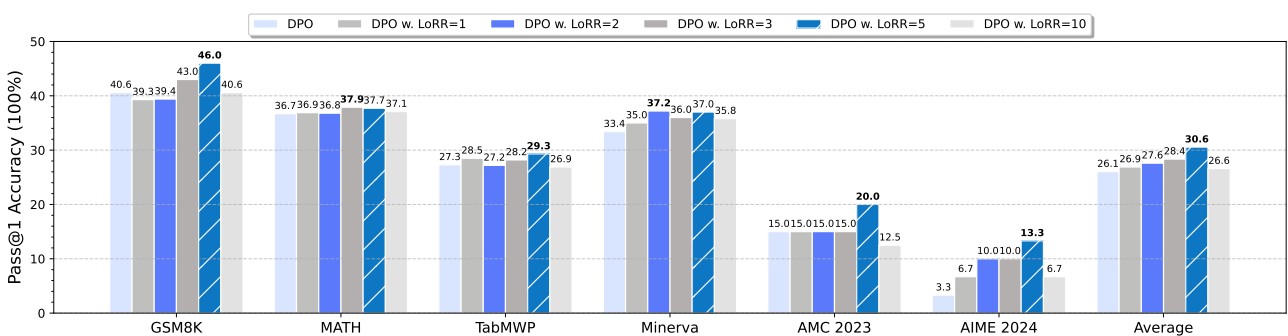

*Figure 6.* The performance of DPO with LoRR on six math tasks under a different replay number. The best results are described in **bold**, and the average of the six tasks is shown at the end.

