# OpenReview forum: "Sample-efficient LLM Optimization with Reset Replay"
_ICML.cc/2026/Conference — Submitted to ICML 2026_

### Official Review · Reviewer_oJHk · 2026-03-09

**Soundness:** 2
**Presentation:** 2
**Significance:** 1
**Originality:** 1
**Overall Recommendation:** 2
**Confidence:** 5

**Summary:**

This paper targets the low sample efficiency and primacy bias issues in preference-based post-training of Large Language Models (LLMs), and proposes LLM Optimization with Reset Replay (LoRR), a plug-and-play plugin module. The core design of LoRR combines four key components: high-replay training to maximize data utility, periodic model reset via the Shrink & Perturb strategy to preserve network plasticity, dynamic mixing of initial offline data and on-policy rollout data (data reset) to mitigate distribution shift, and hybrid optimization with weighted SFT loss and preference loss to stabilize training. The authors conduct experiments on mathematical and general reasoning benchmarks, and report that LoRR brings consistent performance gains to mainstream preference optimization algorithms including DPO, SimPO, KTO, etc.

**Compliance With Llm Reviewing Policy:**

Affirmed.

**Final Justification:**

After carefully reviewing the authors initial rebuttal and subsequent reply to my follow-up comments, I maintain my original recommendation; while the authors addressed some empirical concerns and clarified distinctions between LoRR and concurrent works like SDPO, they failed to resolve core issues of insufficient originality—LoRR remains a combination of existing technologies (data mixing, model resetting, hybrid loss optimization) without a unified theoretical framework, and its dynamic linear scheduling and hybrid optimization strategies lack algorithm-level innovation, merely offering incremental engineering improvements rather than novel scientific or algorithmic contributions, which is insufficient to meet publication standards

**Key Questions For Authors:**

see above

**Limitations:**

see above

**Strengths And Weaknesses:**

# strength
The problem studied in this paper is timely and practically relevant. Low sample efficiency of offline data utilization and catastrophic overfitting caused by primacy bias are critical pain points in the current LLM post-training pipeline, and addressing these challenges has clear value for both academic research and industrial deployment of LLM alignment.

# weakness
The core idea of data reset, the dynamic mixing of initial offline data and on-policy rollout data generated by the current policy, has been extensively studied in LLM continual learning and iterative preference optimization. The paper only designs a simple linear decay schedule for the mixing ratio, without proposing a new data mixing paradigm or distribution shift mitigation mechanism, nor does it provide original modeling and explanation for the data preference differences across different models.

The hybrid optimization of weighted SFT loss and preference loss is an extremely common engineering trick in current LLM preference optimization, and numerous existing works have verified its effect on improving training stability. The paper only designs a linear growth schedule for the SFT weight with replay rounds, without any algorithm-level innovation. recent paper about sft and rl mixing: Reinforcement Learning via Self-Distillation (I think the mixing training idea in the paper is not novel compared with existing work)

Overall, LoRR is essentially a simple splicing and engineering implementation of three independent existing technologies, without forming a unified and original theoretical framework, discovering new scientific phenomena, or proposing a new solution paradigm.

---

> ### Author Rebuttal · Authors · 2026-03-31
>
> We appreciate your time and your recognition that the problem we study—low sample efficiency and primacy bias in LLM post-training—is highly timely and of critical practical relevance.
>
> We understand your concerns regarding algorithmic novelty. **However, we respectfully disagree with the characterization of LoRR as a "simple splicing of independent existing technologies."** Our core scientific contribution lies in identifying the specific vulnerabilities of high-replay preference optimization (namely, severe primacy bias and manifold collapse) and introducing a tightly coupled, synergistic system to resolve them. Below, we address your concerns regarding our design choices and novelty.
>
> ### 1. Synergy, Not "Simple Splicing"
>
> While the broad concepts of data mixing, parameter resetting, and loss mixing exist in deep learning and RL, they have not been systematically unified to unlock the high-replay regime in LLM preference optimization. Crucially, the three components in LoRR are **not independent; they are highly interdependent and mathematically complementary**, as detailed in Line 231.
>
> * Applying high-replay Data Reset alone leads to rapid early stagnation due to primacy bias.
> * Applying Model Reset alone during high-replay leads to severe instability and degeneration, as the newly injected plasticity is easily exploited by the pure preference loss, causing reward hacking.
>
> As clearly demonstrated in our Ablation Study (Figure 5 & Table 3), **only when Model Reset (providing the capacity to learn) is coupled with dynamic Hybrid Optimization (providing the anchor to prevent collapse) can the model survive intensive updates**. Discovering and empirically validating this tightly coupled synergistic mechanism to unlock massive sample efficiency is, we believe, a highly meaningful scientific contribution to the alignment community.
>
> ### 2. Distinction from Existing Hybrid Optimization
>
> We thank you for pointing out the concurrent work, *Reinforcement Learning via Self-Distillation*. We will certainly cite and discuss it in our final version. However, there is a fundamental functional difference between generic hybrid approaches and ours:
>
> * In existing works (like RL via Self-Distillation or standard RLHF), the SFT/NLL loss is typically used with a **static weight** as a continuous, generic regularization term to prevent deviation from the base model.
> * In LoRR, our Hybrid Optimization features a **dynamic growing schedule specifically designed for the late stages of high-replay training**. In early replays, we want maximum preference optimization. However, as the model repeatedly loops over the same data batch, the risk of manifold collapse increases drastically. The linearly growing SFT weight acts as a targeted dynamic stabilizer precisely when the model is most vulnerable. It is an optimization anchor tailored for the "Reset Replay" loop, not a generic static trick.
>
> ### 3. The Value of Simple Linear Schedules
> You noted that our linear decay/growth schedules are "simple." We explicitly chose simple linear schedules over complex, parameterized modeling to ensure LoRR remains a robust, practically useful plug-and-play module. In modern LLM post-training, minimizing computational overhead and hyperparameter tuning is critical. The fact that straightforward linear schedules can robustly manage distribution shifts and bridge offline data with on-policy rollouts across diverse models (e.g., Llama-3, Qwen) and optimization algorithms (DPO, SimPO, KTO) demonstrates the fundamental robustness and generalizability of our framework.
> A more complex dynamic scheduling algorithm (e.g., relying on real-time KL divergence or complex loss landscapes) would introduce excessive computational overhead without guaranteeing proportional performance gains. This can also be left for our future work.
>
> We deeply value theoretical novelty, but we also firmly believe that system-level training frameworks that are elegant, simple, and yield highly robust, sample-efficient empirical improvements (as evidenced by our strong baselines across multiple models) provide immense value to the machine learning community. We will revise these points in our introduction and future work, hoping you might reconsider the value of this synergistic, robust, and highly effective framework.

---

> > ### Author Rebuttal · Reviewer_oJHk · 2026-04-03
> >
> > While the empirical results on specific reasoning tasks are encouraging, the proposed LoRR framework remains a collection of existing heuristics. The claimed 'synergy' is demonstrated only through empirical ablations rather than a rigorous theoretical framework. You definitely can combine several orthogonal tricks to gain the improvement, but it is hard to say you create a novel algorithms, am i right?
> >
> > I can not agree with you about Self-Distillation is "a static weight as a continuous, generic regularization term to prevent deviation from the base model". It actually push the model to align the the reasoning trajectory that yield correct answer. I think comparision with those methods is necessary.
> >
> > Lu, Kevin and Thinking Machines Lab, "On-Policy Distillation"
> > Reinforcement Learning via Self-Distillation

---

> > > ### Author Response · Authors · 2026-04-05
> > >
> > > Thank you for your continued engagement and for taking the time to review our detailed rebuttal.
> > >
> > > Regarding algorithmic novelty and the synergy of our framework, we want to emphasize that the components in LoRR are not merely independent heuristics patched together, but rather interdependent solutions to a specific structural failure mode: the loss of plasticity in high-replay (which may now evolve into self-distillation).  As established in recent deep RL literature [R1, R2], learning on narrow distributions forces networks to lose plasticity (We conducted additional experiments in the rebuttal of Reviewer fhho). Our resets directly resolve this by reviving dead gradients. However, if we only restore plasticity, the unbounded preference loss will instantly exploit this renewed variance to hack the reward model, causing catastrophic forgetting. This is exactly where hybrid optimization comes in as a dynamic anchor.
> > >
> > >
> > > We also appreciate the references to Self-Distillation (e.g., SDPO). **While these works were made arxiv in late Jan 2026 and are classified as concurrent under ICML guidelines (less than two months before the deadline)**, we agree that discussing these paradigms is highly valuable. SDPO is an innovative approach that tackles the spatial credit-assignment bottleneck by utilizing in-context rich feedback to create a 'self-teacher' for dense token-level distillation. Although not token level distillation, we want to highlight that we have already included and outperformed this exact implicit feedback in our main experiments (Table 2, zero-style training e.g., Simple-RL-Zero and ReLIFT).
> > >
> > > LoRR, in contrast, tackles a fundamentally different problem: the temporal primacy bias that occurs when a model must repeatedly replay limited data (or the model self-distillation data) to maximize sample efficiency.
> > > In fact, although LoRR was conducted within the optimization of off-policy like DPO. we believe that the LoRR can also be used with on-policy methods:
> > > SDPO's credit assignment requires rich textual feedback to generate a successful rollout. If one were to apply SDPO's dense credit assignment over multiple high-replay iterations to squeeze maximum utility from a small dataset, the student model would inevitably still suffer from the primacy bias. Our resets could seamlessly integrate into the SDPO pipeline to periodically reset plasticity.
> > > We will explain and include it in the next version for future work.
> > >
> > > [R1] Sokar, et al. The dormant neuron phenomenon in deep reinforcement learning. ICML, 2023.
> > >
> > > [R2] Lyle, et al. Understanding plasticity in neural networks. ICML, 2023.

---

### Official Review · Reviewer_e145 · 2026-03-12

**Soundness:** 3
**Presentation:** 3
**Significance:** 3
**Originality:** 3
**Overall Recommendation:** 5
**Confidence:** 4

**Summary:**

The paper presents a data-efficient approach for improving reasoning in LLMs. The method is largely orthogonal to RLVR, and combines elements of DAPO and SFT to enable more efficient learning. Overall, the empirical results are positive, and the paper includes comprehensive experiments and ablations to support its claims.

**Compliance With Llm Reviewing Policy:**

Affirmed.

**Final Justification:**

The authors resolved my concerns,

**Key Questions For Authors:**

1. The scoring and reward assignment procedure for generated outputs is not sufficiently clear. In particular, the description around L245 would benefit from a more explicit explanation of how generations are evaluated and how the resulting scores are translated into rewards.

2. The discussion in Section 3 around “off-policy” RL-based post-training is also somewhat confusing. My understanding is that RLVR is typically considered an on-policy method, so it would be helpful for the authors to clarify what exactly they mean here. Are they referring specifically to methods such as DPO, or are they using “RL-based post-training” in a broader sense? At present, the paper seems to conflate RL, RLVR, and preference-based post-training

**Limitations:**

Yes

**Strengths And Weaknesses:**

Strengths:
1. The method is simple and elegant. It is well motivated and appears orthogonal to several prior approaches, as also suggested by Table 1
2. The paper is well written and easy to follow.
3. The empirical results are strong across multiple models and datasets, and the paper offers a compelling alternative to RLVR for improving reasoning.
4. The ablation studies are thorough and help support the main design choices.

Weaknesses:
1. The main claim that LoRR is sample efficient is not communicated clearly enough in the main text. A dedicated figure comparing performance as a function of the number of training samples would have made this claim much easier to assess.
2. In some sense, the issue of primacy bias has already been explored in prior post-training work, including DAPO [1], E2H [2], and other curriculum-based approaches. These methods show that explicitly accounting for task difficulty can improve LLM reasoning performance. While this direction is not identical to the present work and is largely orthogonal to it, there is still meaningful conceptual overlap, and the paper would benefit from discussing its relation to these methods.

[1] Yu et al. DAPO: An Open-Source LLM Reinforcement Learning System at Scale. arXiv 2503.14476, NeurIPS 2025, Dec. 2025.

[2] Parashar et al. Curriculum Reinforcement Learning from Easy to Hard Tasks Improves LLM Reasoning. arXiv 2506.06632, Jun. 2025; ICLR 2026.

---

> ### Author Rebuttal · Authors · 2026-03-31
>
> We deeply appreciate your constructive review. It is encouraging that you found our method simple and elegant, and our empirical results strong. We have carefully addressed your feedback below:
>
> ### 1. Clarity of the "Sample Efficient" Claim (W1)
>
> We completely agree that visualizing performance as a function of the number of training samples is the most intuitive way to assess sample efficiency. We would like to respectfully direct your attention to **Figure 4** in our main text, which was designed exactly for this purpose. In Figure 4, we plot the Pass@1 accuracy on the general reasoning benchmark (MMLU-Pro) against the training steps. Since the batch size is constant, the x-axis (steps) directly corresponds to the cumulative number of training samples processed by the model.
> * As illustrated, the performance of LoRR (the blue line) rises significantly faster than all other DPO baselines in the early stages, demonstrating that LoRR extracts more useful learning signals from a smaller number of samples.
> * Moreover, while the baselines struggle to improve or even begin to overfit after consuming a certain amount of samples (around 400 steps), LoRR effectively prevents manifold collapse and continues to gain greater utility, ultimately reaching a much higher performance peak using the same finite dataset.
>
> Furthermore, **Table 1** corroborates this claim from a static dataset perspective: given a strictly identical and limited offline data budget (8K math data), LoRR consistently outperforms all baseline optimizations. We appreciate your valuable feedback. In the revised manuscript, we will update the caption and the text describing Figure 4 to explicitly emphasize that the x-axis reflects the "number of training samples consumed," thereby making our claim of sample efficiency unequivocally clear.
>
>
> ### 2. Relation to Curriculum-based Approaches (W2)
>
> We appreciate you pointing out these highly relevant and recent works (DAPO, E2H). Both lines of work aim to mitigate early-stage learning bottlenecks (such as primacy bias) and improve sample efficiency in LLM reasoning.
>
> As you rightfully noted, these approaches are orthogonal to ours. Curriculum-based methods mitigate primacy bias from a **data perspective** by explicitly ordering the training samples based on task difficulty (from easy to hard). In contrast, LoRR tackles primacy bias from an **optimization dynamics perspective**, utilizing Model and Data resets to restore network plasticity during the high-replay of *any* given data batch. We believe that integrating curriculum data-ordering with LoRR’s reset mechanism could yield even higher sample efficiency. We have added a comprehensive discussion of DAPO and E2H in our Related Work section to properly contextualize our contribution alongside these excellent concurrent efforts.
>
> ### 3. Clarification on Reward Assignment (Q1)
>
> We apologize for the lack of clarity around L245. To clarify, the reward assignments in our framework are driven by a dedicated reward model. Specifically, as mentioned around Line 273, to ensure a fair and standardized comparison, we follow the exact same evaluation protocol as SimPO and utilize `ArmoRM-Llama3-8B` as our reward verifier. The detailed procedure (also described in Appendix A) is as follows:
>
> 1. **Evaluation:** For a given prompt, the current policy generates multiple candidate responses. We then pass these generations through ArmoRM, which evaluates the reasoning and quality, outputting a continuous scalar reward score for each response.
> 2. **Translation to Preferences:** To translate these scalar scores into training signals for preference optimization, we construct preference pairs. Among the generated candidates for a single prompt, the response that receives the highest reward is selected as the "chosen" response ($y_w$), and a response with a lower reward is selected as the "rejected" response ($y_l$).
>
> These explicitly scored preference pairs form the fixed data batch that our LoRR framework subsequently uses for high-replay optimization. We will expand the description around L245 in the revised manuscript to make this scoring and pair construction process completely explicit.
>
>
> ### 4. Terminology in Section 3 (Q2)
>
> Thank you for pointing out the potentially confusing terminology. We fully agree with your understanding: RLVR and PPO are fundamentally *on-policy* methods, as they continuously learn from actively generated rollouts. In contrast, methods like DPO or our LoRR are *off-policy*, as they optimize over (partly) fixed, pre-collected datasets or preference pairs. Our discussion in Section 3 was primarily intended to highlight the vulnerabilities of learning from fixed data batches (i.e., the off-policy setting). We have carefully revised Section 3 to correct this conflation of terms in the next version.

---

> > ### Author Rebuttal · Reviewer_e145 · 2026-04-01
> >
> > The authors solve my concerns, and the paper is relevant in terms of off-policy post-training of LLMs.

---

### Official Review · Reviewer_uWcQ · 2026-03-12

**Soundness:** 3
**Presentation:** 3
**Significance:** 3
**Originality:** 3
**Overall Recommendation:** 4
**Confidence:** 3

**Summary:**

This paper proposes a sample efficient method to make the preference based rl post training more effective. Especially when collecting the new data is expensive. Their method,  LLM optimization with Reset Replay (LoRR), consists of three parts: model reset, data reset, hybrid optimization to make sure the training is stable but not crush. Their experiments show that, across math reasoning tasks. adding LoRR outperforms traditional preference opimization methods.

**Compliance With Llm Reviewing Policy:**

Affirmed.

**Final Justification:**

The author resolved my concerns with additional study. I will raise my score.

**Key Questions For Authors:**

- I think the most critical question would be the KL one
- I might need explnations or experiments for the math only benchmarks

**Limitations:**

An ablation study for KL.

**Strengths And Weaknesses:**

Strength:
- Clear writing. The paper clearly mentioned their motivation and experiment design.
- Strong baselines. The paper shows the improvements across different preference optimization methods

Weakeness:
- If the goal of the model reset is to prevent the policy from diverging too much from the initial policy model, the most straightforward approach is to increase the KL coefficient. I think it needs to be validated with an experiment.
- Benchmarks only contain math questions. Some commonsense reasoning tasks might be helpful.
- The paper claimed that hybrid optimization is helpful to prevent manifold collapse, is this collapse a kind of entropy too small or just generating meangingless sentences?

---

> ### Author Rebuttal · Authors · 2026-03-31
>
> We sincerely thank you for your constructive feedback. We have carefully addressed your insightful questions below:
>
> ## Model Reset vs. KL (W1 & Q1)
>
> We would like to clarify that the primary objective of Model Reset is **NOT merely to prevent divergence, but to restore network plasticity and mitigate primacy bias** during high-replay training.
>
> While increasing KL (e.g., $\beta$ in DPO) acts as a strict distributional constraint forcing the policy, it introduces a severe trade-off in a high-replay setting. Specifically, a large KL penalty overly restricts the optimization space, which severely limits the model's ability to learn and leads to underfitting or early stagnation. In contrast, our Model Reset directly on the weight space periodically. This effectively breaks "dead neurons" and recovers the model's plasticity, enabling the network to continuously extract new utility from offline data without overfitting.
>
> To empirically validate this, we scale the KL penalty ($\beta \in \{0.01, 0.1, 0.5, 1.0\}$) using standard DPO, and compared it against LoRR. The results are presented below:
>
> | Model | gsm8k | math | tabmwp | minerva_math | amc23 | aime24 | avg |
> |---|---|---|---|---|---|---|---|
> | Llama 3B | 37.7 | 35.1 | 27.5 | 33.8 | 12.5 | 6.7 | 25.55 |
> | DPO_0.01KL | 40.6 | 36.7 | 27.3 | 33.4 | 15.0 | 3.3 | 26.1 |
> | DPO_0.1KL | 37.5 | 36.1 | 27.4 | 35.4 | 20.0 | 6.7 | 27.2 |
> | DPO_0.5KL | 38.2 | 35.4 | 27.4 | 34.6 | **22.5** | 6.7 | 27.5 |
> | DPO_1.0KL | 37.8 | 36.5 | 26.6 | 35.8 | 17.5 | **10.0** | 27.4 |
> | **LoRR (Llama)** | **43.0** | **37.9** | **28.2** | **36.0** | 15.0 | **10.0** | **28.4** |
> | Qwen 3B | 41.1 | 60.4 | 32.1 | 60.8 | 25.0 | 3.3 | 37.1 |
> | DPO_0.01KL | 44.3 | 60.9 | 32.4 | 60.2 | 25.0 | 0.0 | 37.1 |
> | DPO_0.1KL | 45.0 | 60.8 | 30.5 | 60.8 | 32.5 | **6.7** | 39.4 |
> | DPO_0.5KL | 44.3 | 60.8 | 31.6 | 61.0 | 27.5 | 3.3 | 38.1 |
> | DPO_1.0KL | 45.3 | 60.7 | **32.8** | **62.4** | 30.0 | **6.7** | 39.6 |
> | **LoRR (Qwen)** | **45.4** | **61.2** | 32.4 | 61.2 | **32.5** | **6.7** | **39.9** |
>
> As shown in the table, merely increasing the KL penalty ultimately stagnates. Conversely, LoRR successfully maintains plasticity and achieves the highest average performance across both Llama and Qwen models. We have added this ablation study to our revised manuscript.
>
> ## Evaluation on Reasoning Tasks (W2&Q2)
>
> We completely agree that evaluating reasoning tasks is crucial for demonstrating the generalizability of our method. While we had included preliminary explorations on general domains in Figure 4 and Line 316 of our original submission (showing gains on MMLU-Pro), we have now conducted extensive additional experiments on 9 widely-used QA benchmarks (NQ, HotpotQA, SimpleQA, etc.).
>
> As demonstrated in the tables below, **LoRR consistently outperforms the pure DPO baseline across all evaluated commonsense tasks** in F1 metrics:
>
> | Model | nq | triviaqa | popqa | hotpotqa | 2wikimultihopqa | musique | bamboogle | facts | simpleqa | avg |
> |---|---|---|---|---|---|---|---|---|---|---|
> | gpt-4o-2024-11-20 | 0.4274 | 0.3937 | 0.4331 | 0.4246 | 0.4231 | 0.4604 | 0.4730 | 0.4659 | 0.4527 | 0.4393 |
> | llama | 0.1475 | 0.1279 | 0.1590 | 0.1544 | 0.1597 | 0.1721 | 0.1739 | 0.1711 | 0.1711 | 0.1596 |
> | llama_dpo | 0.1803 | 0.1623 | 0.1951 | 0.1912 | 0.2000 | 0.2089 | 0.2146 | 0.2109 | 0.2096 | 0.1970 |
> | **llama_lorr** | **0.1824** | **0.1656** | **0.2002** | **0.1947** | **0.2024** | **0.2159** | **0.2200** | **0.2161** | **0.2147** | **0.2013** |
> | qwen | 0.1718 | 0.1428 | 0.1717 | 0.1650 | 0.1646 | 0.1828 | 0.1847 | 0.1852 | 0.1864 | 0.1728 |
> | qwen_dpo | 0.1683 | 0.1367 | 0.1702 | 0.1631 | 0.1646 | 0.1803 | 0.1817 | 0.1826 | 0.1831 | 0.1701 |
> | **qwen_lorr** | **0.1720** | **0.1408** | **0.1740** | **0.1655** | **0.1661** | **0.1840** | **0.1850** | **0.1857** | **0.1859** | **0.1732** |
>
> These results validate that LoRR's efficiency and performance gains are not limited to math domains, but are broadly applicable to reasoning tasks. We will include both EM and F1 results in the next version.
>
> ## Clarification on "Manifold Collapse" (W3)
>
> By "manifold collapse" in the context of LLM preference optimization, we primarily refer to the phenomenon where the model generates meaningless, grammatically broken sentences, or repetitive gibberish. During intensive high-replay updates, the optimizer tends to over-optimize the specific preference/reward signal, pushing the latent representations off the natural language manifold. While this is sometimes accompanied by a sharp drop in generation entropy (mode collapse), the practical and severe manifestation is the generation of meaningless text. Our Hybrid Optimization, which dynamically mixes the CE loss for language capacity, acts as a regularization anchor. It ensures that while the model learns to maximize preferences, its outputs remain firmly grounded in the natural language distribution. We have clarified this definition in Section 4 of the revised manuscript.

---

> > ### Author Rebuttal · Reviewer_uWcQ · 2026-04-02
> >
> > The author resolved my concerns. I will raise the score.

---

### Official Review · Reviewer_fhho · 2026-04-03

**Soundness:** 3
**Presentation:** 3
**Significance:** 3
**Originality:** 2
**Overall Recommendation:** 4
**Confidence:** 4

**Summary:**

This paper proposed LLM Optimization with Reset Replay (LoRR), enhancing the sample efficiency through dataset reset, and incorporating the model reset and a training method of hybrid optimization to improve performance. Extensive experiments on different models and optimization methods show the effectiveness.

**Compliance With Llm Reviewing Policy:**

Affirmed.

**Final Justification:**

This paper is simple and easy to follow. By proposing LLM Optimization with Reset Replay (LoRR), they enhance the sample efficiency through dataset reset, and incorporates the model reset and a training method of hybrid optimization to improve performance. Extensive experiments on different models and optimization methods show the effectiveness on the math benchmark. Though there may be some concerns regarding that no new algorithm is proposed, but a combination of existing methods, I think it's quite novel to introduce the "Reset", "plasticity" into LLM optimization. Therefore, I recommend this paper to weak accept.

**Key Questions For Authors:**

I have listed the questions in the weakness part, so you can refer to it.

**Limitations:**

Yes.

**Strengths And Weaknesses:**

**Strength**
1. This paper is easy to follow and well-written.
2. The method is simple but effective. No complex new methods are introduced; instead, the problem is solved through a combination of existing simple approaches, making it intuitive and concise.
3. Experiments were conducted across multiple models and optimization algorithms, and the experimental analysis is fairly thorough. The experimental results on the math benchmarks are consistent across vast of algorithm, such as KTO, DPO.


**Weakness**
1. Comparison results in Tables 1, 5, and 6: since your method uses data replay, does it train on more data per step than the baseline, or perform more gradient optimization steps?
2. The paper mentions “plasticity,” but I don’t seem to see any analysis based on plasticity-related metrics. Is improved performance directly taken as evidence that plasticity is preserved?
3. For the component ablation study, only data reset and hybrid optimization are included, but there are no results for model reset. Additionally, I’m curious about the effects of pairwise combinations of these mechanisms.

---

### Decision · Program_Chairs · 2026-04-30

**Decision:**

Reject

**Comment:**

Update for final decision: After calibrating with the SAC, we determined that this paper is just short of the acceptance bar. I encourage the authors to make the edits in the discussion and resubmit.

Reviewers generally agreed that the introduced method, LLM optimization with Reset Replay, was empirically demonstrated to be effective through a good selection of ablations and across several models. While the method is a combination of several in the literature, the combination is novel and well explained. However, there was not a consensus among reviewers, as one reviewer was strongly against publication, citing a lack of novelty as the main concern; the reviewer claimed that the paper merely stitched a few well-known ideas together. In response, the authors point out that all three components are synergistic and ablations show they are all necessary to obtain the performance gains. Reading through the reviewers and looking at the paper, I agree with the authors and recommend acceptance despite the dissenting review.